# Impact of Pre-Existing Immunity and Age on Antibody Responses to Live Attenuated Influenza Vaccine

**DOI:** 10.3390/vaccines12080864

**Published:** 2024-08-01

**Authors:** Lukas Hoen, Sarah Lartey, Fan Zhou, Rishi D. Pathirana, Florian Krammer, Kristin G -I Mohn, Rebecca J. Cox, Karl A. Brokstad

**Affiliations:** 1Influenza Centre, Department of Clinical Science, University of Bergen, Haukelandsbakken, 5009 Bergen, Norway; sarah.lartey@uib.no (S.L.); fan.zhou@uib.no (F.Z.); rishi.pathirana@gmail.com (R.D.P.); kristin.mohn@uib.no (K.G.-I.M.); rebecca.cox@uib.no (R.J.C.); karl.brokstad@uib.no (K.A.B.); 2Department of Microbiology, Icahn School of Medicine at Mount Sinai, New York, NY 10029, USA; florian.krammer@mssm.edu; 3Center for Vaccine Research and Pandemic Preparedness (C-VaRPP), Icahn School of Medicine at Mount Sinai, New York, NY 10029, USA; 4Department of Pathology, Molecular and Cell Based Medicine, Icahn School of Medicine at Mount Sinai, New York, NY 10029, USA; 5Ignaz Semmelweis Institute, Interuniversity Institute for Infection Research, Medical University of Vienna, 1090 Vienna, Austria; 6Department of Medicine, Haukeland University Hospital, 5021 Bergen, Norway; 7Department of Microbiology, Haukeland University Hospital, 5021 Bergen, Norway; 8Department of Safety, Chemistry and Biomedical Laboratory Sciences, Western Norway University of Applied Sciences (HVL), 5063 Bergen, Norway

**Keywords:** live attenuated influenza vaccine, immune response, functional antibodies, children, adults

## Abstract

Live attenuated influenza vaccines (LAIV) typically induce a poor hemagglutination inhibition (HI) response, which is the standard correlate of protection for inactivated influenza vaccines. The significance of the HI response is complicated because the LAIV vaccine primarily induces the local mucosal immune system, while the HI assay measures the circulating serum antibody response. However, age and pre-existing immunity have been identified as important factors affecting LAIV immunogenicity. This study aimed to extend our understanding of LAIV-induced immunity, particularly, the impact age and pre-existing immunity have on eliciting functional and neutralising antibody responses in paediatric and adult populations vaccinated with LAIV. Thirty-one children and 26 adults were immunized with the trivalent LAIV during the 2013–2014 influenza season in Norway. Children under 9 years received a second dose of LAIV 28 days after the first dose. Blood samples were collected pre- and post-vaccination. HI, microneutralization (MN) and enzyme-linked lectin assay for neuraminidase (NA) antibodies were measured against the vaccine strains. IgG antibody avidity against hemagglutinin (HA) and NA proteins from the vaccine strains was also assessed. Significant correlations were observed between HI and MN responses to A/California/7/2009 (A/H1N1)pdm09-like strain and B/Massachusetts/2/2012-like strain, suggesting that MN is a potential immunological correlate for LAIV. However, the relationship between recipient age (or priming status) and serological response varied between vaccine strains. There was a notable increase in HI and MN responses in all cohorts except naive children against the H1N1 strain, where most recipients had responses below the protective antibody threshold. NAI responses were generally weak in naive children against all vaccine strains compared with adults or antigen-primed children. Post-vaccination antibody avidity increased only in primed children below nine years of age against the A/H1N1 strain. Overall, our findings indicate that LAIV elicits functional and neutralizing antibody responses in both naive and antigen experienced cohorts, however, the magnitude and kinetics of the response varies between vaccine strains.

## 1. Introduction

Influenza viruses are a major cause of global respiratory illness that results in 290,000–646,000 fatalities and approximately 3–5 million hospitalisations annually [1]. Vaccination remains the most effective countermeasure against influenza, with three vaccine types currently in use: inactivated influenza vaccines (IIV), recombinant haemagglutinin protein, and live attenuated influenza vaccines (LAIV). LAIV is approved in Europe for children between the ages of 2–17 years; in the USA, it is approved for a wider age range between 2 and 49 years [2,3]. Prior to the introduction of the 2009 H1N1 pandemic strain into human circulation, LAIV demonstrated significantly higher efficacy in paediatric cohorts [4,5,6]. In contrast, the opposite was true for adults, suggesting age and pre-existing immunity may interfere with the mechanism of action of LAIV [7,8,9]. The assessment of LAIV induced immunogenicity in clinical trials is complicated because there are currently no clearly defined correlates of protection (COP) for this vaccine [10]. The hemagglutination inhibition (HI) assay measures neutralizing antibodies targeting the receptor binding site on Hemagglutinin (HA) and an HI titre of ≥40 is accepted as a COP for seasonal inactivated and recombinant haemagglutinin influenza vaccines by regulatory agencies [11,12]. HA is the most dominant external glycoprotein on influenza virions, and antibodies capable of sterically hindering interactions with host sialic acids may prevent virion internalization. However, the HI assay underestimates the level of protection offered by LAIV vaccination, likely due to the multifaceted immune response that contributes to its mechanism of action [13]. LAIV mimics natural infection with limited viral replication, eliciting local IgA and T cell responses not measured by the HI assay [14]. Furthermore, the level of pre-existing immunity against influenza and the recipient’s age are key determinants of the level of protective immune responses induced by LAIV. Pre-existing immunity gained through prior vaccination and/ or natural infection limits the replication of LAIV in adults, thus reducing its immunogenicity in this cohort [15]. We have previously reported that naïve children induced a rapid protective antibody response (14 days) after LAIV vaccination, while antibody, memory B cell and antibody-secreting cell levels were not significantly boosted in primed children (HI titres ≥ 40) [16]. Therefore, when assessing the immunogenicity of LAIV, factors such as priming status, age, and functional antibody responses must be considered. 

A potential COP for LAIV is to measure enzymatically inhibiting antibody responses against neuraminidase (NA). NA is responsible for hydrolysing terminal sialic residues from N-linked glycans, allowing the virus to abrogate any interactions with defensive proteins and releasing newly formed virions by accumulating at the host cell membrane [17]. NA-specific antibodies capable of inhibiting enzymatic activity have been shown to reduce the duration and severity of influenza infections [18]. The quality of the immune response following vaccination can be further assessed by measuring antibody avidity, which is indicative of the induction of immunological memory [19]. Avidity maturation in response to inactivated seasonal influenza vaccination is highly variable because of previous antigen exposure and age [20,21]. Higher avidity against HA has been shown to correlate with increased protection against the A/H1N1 2009 pandemic strain in elderly individuals [21,22]. Measurement of antibody avidity, therefore, provides insight into the generation of protective and long-term antibody and B cell memory responses. The microneutralization (MN) assay is another widely used serological assay for detecting infection or vaccination-induced functional neutralizing antibody responses. Here we used an MN assay that quantifies an antibody concentration correlated to an optical density value measured using an anti-nucleoprotein immuno-staining in a permissive cell line in vitro [23]. In contrast to the HI assay, the MN is highly sensitive and can also be used to assess functional antibodies, thus providing an alternative approach to assessing influenza vaccine immunogenicity and a potential COP. 

This study aims to broaden our understanding of the factors that influence vaccine immunogenicity and identify a potential surrogate marker of protection of LAIV. Here we compared functional neutralizing antibody responses and antibody avidity in a cohort of children (aged between 3 and 17) and adults after LAIV vaccination. Data presented here show a highly variable serological response to LAIV dependent on pre-existing immunity and the vaccine strain.

## 2. Materials and Methods

### 2.1. Study Design and Sampling 

During the October 2013 to February 2014 influenza season, children (*n* = 31) and adults (*n* = 26) scheduled for tonsillectomy at Haukeland University Hospital were recruited and vaccinated with trivalent LAIV. The patients were scheduled for elective tonsillectomy due to chronic tonsillitis, tonsillar hypertrophy, or both but were otherwise healthy. A potential bias in cohort selection may stem from adults who allow their children to receive seasonal influenza vaccinations and participate in studies requiring multiple checkups. Although participants recruited underwent a tonsillectomy, we have previously shown that this procedure does not impact total serum antibody levels [24]. This was a collaborative vaccine trial between the Influenza Centre, the Paediatric clinical trial unit, and the Ear, Nose, and Throat outpatient clinic at Haukeland University Hospital (Bergen, Norway). All adults and the children’s guardians provided written informed consent before joining the study. The inclusion criteria for the clinical trial were no influenza-like illness 7 days before vaccination. Females of childbearing age required a negative pregnancy test and participants with mild-to-moderate asthma were included. The exclusion criteria comprised children and adults with chronic medical conditions, pregnant women, patients taking acetylsalicylic acid, or immunosuppressants. The clinical trial was approved by the Regional Ethical Committee of Western-Norway (REC West 2012/1008), the Norwegian Medicines Control Agency, and was registered in the European Clinical Trials Database (EUDRACT 2012_002848-24) and National Institute for Health Database (www.clinicaltrials.gov; (NCT01866540). All adults and children older than 9 years old (*n* = 7) received one dose of LAIV at day 0, and children under 9 years old (*n* = 24) received a second dose at day 28 according to the manufacturer’s recommendation. Blood samples were collected prior to vaccination (day 0), median day of tonsillectomy (3–22 days), 28 days, and 56 days post-vaccination. Plasma samples were separated from a heparin blood sample and stored at −80 °C until further used in the serological assays.

### 2.2. Live Attenuated Influenza Vaccine (LAIV)

The vaccine used in the study was Fluenz (AstraZeneca, Cambridge, UK), which contained 107.0 ± 0.5 fluorescent focus units (FFU) of three live attenuated influenza viruses; A/California/7/2009 (A/H1N1) pdm09-like strain (MEDI 228029), A/Texas/50/2012 (A/H3N2)-like strain (MEDI 237514), and B/Massachusetts/2/2012-like strain (MEDI 23775). The vaccine was administered (according to the manufacturer’s recommendation) intranasally with a 0.2 mL dose per nostril. None of the participants had been previously vaccinated with LAIV.

### 2.3. Hemagglutination Inhibition (HI) Assay 

HI results have been previously published [14]. Briefly, 0.7% turkey red blood cells were incubated with plasma treated with receptor destroying enzyme, and subsequently incubated with 8 hemagglutinating units of A/California/7/2009 (pdm09 H1N1), A/Texas/50/2012 (A/H3N2), or B/Massachusetts/2/2012 virus. Undetectable titres were assigned a value of 5, and HI titres were defined as reciprocal the maximum dilution of plasma capable of inhibiting 50% of hemagglutination.

### 2.4. Microneutralization Assay 

A modified microneutralization assay was performed in this study [25]. Briefly, plasma samples were heat-inactivated at 56 °C for 30 min. A two-fold plasma dilution series was carried out starting from 1:10 dilution and subsequently incubated with live A/California/7/2009 (pdm09 H1N1), A/Texas/50/2012 (A/H3N2), or B/Massachusetts/2/2012 live viruses for 1 h. A total of 100 μL Madin Darby canine kidney (MDCK) cells (1.5 × 105/mL) were added to each well and incubated for 18 h at 37 °C with 5% CO_2_. Following the overnight incubation period, the cells were fixed in methanol containing 0.6% H_2_O for 20 min at room temperature. Primary antibodies specific to each viral subtype were used at a 1:3000 dilution: mouse anti-influenza A nucleoprotein (H1N1), anti-influenza A (Clone A3) (A/H3N2), and mouse anti-influenza B nucleoprotein (influenza B), were incubated for 1 h at 37 °C. Secondary antibodies were added to each well for a 1-h incubation at 37 °C, with horseradish peroxidase-conjugated polyclonal rabbit anti-mouse IgG (Agilent-Dako #P0260, Santa Clara, CA, USA) at a 1:7500 dilution, specific for both influenza A virus subtypes, and influenza B virus was incubated with HRP conjugated goat anti-mouse IgG (Invitrogen #31430, Waltham, MA, USA) at a 1:10,000 dilution. The plates were incubated for 18 min in darkness with TMB (3,3′,5,5′-tetramethylbenzidine) (BD Biosciences #555214, San Diego, CA, USA). The reaction was stopped with 100 mL of 0.5 M HCl. Plates were then read immediately at 450 nm, with 620 nm reference on the enzyme-linked immunosorbent assay (ELISA) plate reader (Synergy hybrid reader: Biotek), to obtain an optical density value. The MN titre was defined according to curve fitting the reciprocal of the highest plasma dilution capable of inhibiting 50% of virus infectivity. 

### 2.5. ELLA 

The enzyme-linked lectin assay (ELLA) was used to quantify NA inhibiting (NI) antibody titres after LAIV vaccination as previously described [26]. In this study the following viruses were used as the source of NA, A/Equine/Prague/56 (H7) + A/California/07/09 (N1), A/turkey/Massachusetts/3740/1965 (H6) + A/Texas750/2012 (N2), and A/turkey/Massachusetts/3740/1965 (H6) + B/Yamagata/16/1988 (NB). Briefly, 96 well microtiter plates (Thermo Fischer Scientific MaxiSorp cat# 5150-0014, Waltham, MA, USA) were coated with 25 μg/mL of fetuin (Sigma-Aldrich, F3385, Burlington, MA, USA) dissolved in 1× coating buffer (KPL-Sera lab) for 18 h at 4 °C. Plates were washed 3 times with phosphate buffer solution (PBS)-Tween 20 (0.05%). A 5-fold serial dilution of the plasma was carried out before adding 50 μL of a standard amount of virus. The plates were then incubated for 20 h at 37 °C with 5% CO_2_. After overnight incubation, the plates were washed six times with PBS-T and then incubated with 100 μL/well of HRP conjugated lectin from *Arachis hypogaea* (Sigma cat# L7759) solution for 2 h at room temperature in the dark. Plates were washed 3 times, then 100 μL of OPD (o-phenylenediamine dihydrochloride (Sigma cat# P8287) substrate was added and then incubated for 10 min at room temperature in the dark. The reaction was stopped by adding 100 μL/well of 1 N sulfuric acid. The optical density (OD) was measured at 490 nm. The neuraminidase inhibition titre was then defined as the 50% inhibitory concentration using a sigmoidal dose response curve. 

### 2.6. Avidity ELISA

Prior to performing the avidity ELISA, the antibody endpoint titre against HA and NA was determined by ELISA. Briefly, 96 well microtiter plates (Maxisorp plates Cat: 10394751) were coated overnight with either HA or NA at 1 μg/mL in PBS at 4 °C. Subsequently, plates were washed 6 times in PBS + 0.05% Tween 20 and then blocked with blocking buffer (5% milk powder, 0.1% Tween 20, 1% bovine serum albumin) for 1 h at 37 °C. Then, plasma was diluted in a blocking buffer 5-fold serial dilution. Plates were washed as previously stated and incubated with monoclonal mouse anti-human IgG HRP 1:4000 dilution in a blocking buffer. After washing, TMB reagents A and B were mixed at a 1:1 ratio directly prior to their addition to plates. A room-temperature incubation in the dark was carried out for 10 min. The reaction was stopped with 100 μL/well of 0.5 M HCl, and the absorbance was read at 450 nm with a reference of 620 nm. Avidity ELISA was then carried out with plasma diluted to the appropriate OD value of 0.7 ± 0.3 in blocking buffer and incubated for 1 h at 37 °C. Subsequently, a PBS-T wash step was carried out. Sodium thiocyanate (NaSCN) was diluted in PBST (0.3%) to 1.5 M (treated) or 0 M (non-treated), then added to every other row and incubated for 1 h at 37 °C. NaSCN was removed and washed with 0.05% PBS-Tween 20 using a vacuum pump. Monoclonal mouse anti-human IgG HRP was then added at a 1:4000 dilution for a 1-h incubation period. Plates were washed as previously described, TMB reagents were added and incubated for 18 min in the dark. The reaction was then stopped with 100 μL/well with 0.5 M HCl, and the absorbance was read at 450 nm with a reference of 620 nm as previously described.

### 2.7. Statistical Analysis 

Geometric means were used to analyse the non-normally distributed data. Statistical significance was assessed using the Friedman test due to repeated measures of participants and small sample sizes. Post-hoc analysis of individual groups was conducted using the Dunn test. The 95% confidence intervals were displayed to visualize the precision of the data. Deming regression was used to account for measurement errors when correlating two variables that are not independent. Spearman’s R correlation summarized the non-parametric nature of our dataset. All children were grouped together for the Spearman’s R correlation to adjust for sample size. Exact *p*-values are available in Appendix A. A minimum of 2 measured time points were required for inclusion in Section 3, and on average, only 3 individuals were missing 1 data time point per assay.

## 3. Results

### 3.1. Cohort Stratification

LAIV was well tolerated in both paediatric and adult cohorts [14]. Full age group analysis showed an increase in HI, MN, and NAI responses after vaccination with LAIV (Appendix A). To further understand the role age and pre-existing immunity have on LAIV-elicited immunogenicity, we stratified our cohort according to the number of LAIV doses (age), and pre-existing immunity in young children under the age of 9 years.

Children under the age of nine received two doses of LAIV (on days 0 and 28) (Table 1). This group was divided based on their pre-vaccination strain-specific baseline HI titre. Those with HI titre ≤ 5 were classified as “naive children < 9 years”, and those with detectable HI titres of >5 were classified as “primed children ≤ 9 years”. The third group, “primed children > 9 years”, and the fourth group, “Adults” > 18, received only one dose of LAIV at day 0.

### 3.2. Hemagglutination Inhibition Responses to LAIV Vaccination

An HI titre of ≥40 is the accepted correlate of protection for inactivated influenza vaccines [12]. Here we find that prior to vaccination, no HI titres were detected against A/H1N1 in naive children, with a geometric mean titre (GMT) of 5.0 Figure 1A). In adults, LAIV-elicited H1N1-specific HI titres above the protective threshold of 28 days post-vaccination (GMT 48.8) with 69% of vaccinees having HI titres above the protective level of 40. In contrast, only two naive children had HI titres ≥ 40 at 28 days post-vaccination. The HI GMT titres of primed children < 9 years and primed > 9 years increased from 58.9 and 49.9 prior to vaccination to 89.6 and 135.4 56 days post-vaccination, respectively. At day 56, 78% of primed children < 9 years and 86% primed children > 9 years and 69% of adults had protective HI titres ≥ 40. 

A majority (71%) of primed children < 9 years had HI titres against A/H3N2 prior to vaccination (GMT 123.7) and this increased to 80% at 56 days post-vaccination (GMT 174.7) (Figure 1B). In contrast, both naive children < 9 years and primed children > 9 years had low pre-existing HI antibody titres against A/H3N2, however, a majority (63% and 86%, respectively) elicited protective titres at 56 days post-vaccination. In adults, 35% had protective HI titres against A/H3N2 prior to vaccination (GMT 21.6) and the number 7 of individuals with protective titres increased to 46% by day 56 post-vaccination. 

A majority of the primed children < 9 years (67%, GMT 59.2), >9 years (57%, GMT 59.0) and adults (81%, GMT 78.7) had protective HI titres against the B/Mass strain prior to vaccination. By day 56, all vaccinees in these three groups elicited protective HI responses to the B/Mass virus. In naive children, there was a significant increase in the HI GMT (136.0) and the number of vaccinees with protective titres (89%) by day 28 post-vaccination, compared with pre-vaccination levels (0% protective and GMT 5.0). At day 56 post-vaccination, all the naive children had protective HI titres against the B/Mass strain.

### 3.3. Neutralising Antibody Response to LAIV Vaccination

Currently, MN titres are not a COP accepted by regulatory agencies [12]. However, a previous study [27] suggests that an MN titre of 80 correlates with an HI titre of 40 and was used as a benchmark value. LAIV vaccination failed to induce a significant MN response against the H1N1 strain in naive children, with only two vaccinees having MN titres above 80 at day 56 post-vaccination (Figure 2A). In contrast, 45% of primed children < 9 years, had titres above 80 pre-vaccination (GMT 52.9) and this number increased to 78% at day 56 post-vaccination (GMT 109.0). Similarly, the MN response in primed children > 9 years increased from 57% having titres above 80 at baseline (GMT 74.5) to 83% at day 28 and 86% at day 56. In adults, most subjects (58%, GMT 76.2) had MN titres against H1N1 above 80 prior to vaccination and this increased slightly to 62% (GMT 85.0) at day 7 and plateaued until day 56 post-vaccination.

A majority of individuals had MN titres above 80 against the A/H3N2 strain prior to vaccination (Figure 2B). We observed statistically significant increases in MN GMTs against A/H3N2 in primed children < 9 years and >9 years, and adults following vaccination. In naive children, we observed a trend of increase in MN GMT titres against A/H3N2 after LAIV vaccination, although this was not statistically significant. 

The MN responses elicited against the B/Mass strain were in general lower than that observed against the A/H1N1 and A/H3N2 strains. At baseline, a majority of participants in all groups had MN titres less than 80 (Figure 2C). The highest fold-change in the MN response was observed in naive children, where the GMT MN titre increase from 17.7 at day 0 to 115.9 on day 56, with 56% of naive children eliciting MN titres above 80 at this timepoint. Fold-changes for all data are specified in Appendix A. Following vaccination, the number of vaccinees with MN titres above 80 against B/Mass also increased among primed children, albeit the GMT fold-change at day 56 was less than that observed in the naive children. In adults, there was a significant increase in B/Mass-specific GMT MN titre, however, only 42% of individuals elicited MN titres above 80 at 56 days post-vaccination. 

### 3.4. Correlation between Hemagglutination Inhibition and Microneutralizing Antibody Responses 

We assessed whether there was a correlation between the influenza-specific HI and MN antibody responses in children and adults following LAIV vaccination. A Deming regression analysis method was used to account for the measurement error in both X (HI) and Y (MN) titres in Figure 3. Accordingly, a slope closer to 1 indicates a closer correlation. Point estimate slopes against A/H1N1 and B/Massachusetts ranged from 0.9664 to 1.462 in children and adults, thus indicating the correlations between HI and MN against these two strains were statistically significant. A similar significant correlation between HI and MN responses against A/H3N2 in adults was observed, albeit with a point estimate slope of 0.6912. In contrast, MN and HI responses against A/H3N2 in children were non-significantly correlated. 

### 3.5. Neuraminidase Inhibiting Antibodies

Next, we assessed the neuraminidase inhibiting (NAI) antibody response following LAIV, which has previously been shown to limit disease severity [28,29]. Here we used the neuraminidase genotype of interest, and a haemagglutinin genotype which our cohort is unlikely to have been exposed to previously. 

Here we show that pre-existing GMT A/N1 specific NAI antibody titres vary between the different cohorts at baseline but increase following LAIV (Figure 4A). We observed a statistically significant increase in GMT NAI antibody titres against A/N1 in adults as early as 7 days after vaccination and a trend of increasing NAI antibody titres in naive, primed < 9 years, and primed > 9 years children at day 28 after LAIV vaccination. In contrast to A/N1, the GMT A/N2 specific NAI antibody titres were consistently higher at baseline in all cohorts tested. Following vaccination, modest increases in GMT HAI titres were observed against the A/N2 strain, however, none reached statistical significance compared with baseline levels. 

NAI titres against the B virus were measured with neuraminidase (B/Yamagata/16/1988) (NB). The NAI responses against the NB at baseline were low in naive (GMT 5.4) and primed children < 9 years old (GMT 6.0) compared with primed children > 9 years (GMT 23.1) (Figure 4C). The adult cohort elicited the highest baseline NAI response against NB/Yamagata strain (GMT 117.5). Our findings show a significant increase in NB-specific NAI antibody titres in the primed children < 9 years group from 5.5 on day 0 to 10.4 at 56 days after vaccination. The GMT NAI responses in the naive and primed children < 9 years cohorts also increased on days 28 and 56 compared with baseline levels, albeit not significantly. In contrast, the adult NAI titres against the NB strain did not change significantly following vaccination but remained at higher levels compared to the other cohorts.

### 3.6. Antibody Avidity against Haemagglutinin and Neuraminidase Following LAIV Vaccination 

Avidity index values are expressed as a percentage of bound IgG following treatment with 1.5 M NaSCN (Figure 5 and Figure 6). In general, we did not observe any significant changes in antibody avidity following LAIV vaccination, except for the avidity against A/H1N1 HA in the primed children < 9 years old cohort. In this group, a significant increase in the avidity was observed at day 28 (13.7% avidity index) compared with baseline levels (9.4% avidity index), and the avidity was increased further at 56 days post-vaccination (17.7% avidity index). The lowest levels of antibody avidity against HA were observed in the naive children cohort against the A/H1N1 and B/Mass strains. In contrast, the highest levels of antibody avidity against HA were detected in the adult cohort against the A/H3N2 strain, however, no significant increase in avidity was observed following vaccination. 

We did not observe any significant changes in antibody avidity against NA following LAIV vaccination (Figure 6). A modest increase in antibody avidity against the NA from the three vaccine strains was observed in the adult cohort after vaccination, however, this was not statistically significant.

### 3.7. Cross Correlation of Immunological Responses in Children and Adults Following LAIV Vaccination 

LAIV-elicited immunological responses were correlated using Spearman R analysis to identify potential relationships between different antibody responses (Figure 7). Data was normalized by taking the fold change between day 0 and day 56 after vaccination and then averaged across all three vaccine strains. Rank correlation values closer to 1 indicate a positive correlation between two variables. Rank correlation values closer to –1 indicate an inverse correlation between two variables. A rank correlation value approaching zero indicates that there is no apparent correlation between the two variables. 

The most significant positive correlation was between MN and NAI in children at 0.67 (*P*: 4.012 × 10^−5)^ (Figure 7). The only other significant rank correlation values were between HI and MN, at 0.48 in children (*P*: 0.006), and 0.61 in adults (*P*: 0.001). 

## 4. Discussion

A crucial aspect in assessing vaccine immunogenicity is the establishment of correlates of protection. To date, the HI titre and single radial haemolysis are the only COPs accepted by regulatory agencies. However, a challenge with using HI titres as a measure of protection is that LAIV, compared to IIVs, is a poor inducer of HI antibody responses [30]. Despite the relatively poor induction of HI antibody responses, LAIV has demonstrated superior protection in children compared to IIV [31]. In earlier studies, we reported the mucosal and systemic immune response induced by LAIV in paediatric and adult populations [14]. Our current study extends our analysis to encompass various functional antibody responses and antibody affinity maturation in the same cohorts. 

A significant finding in this study is the positive correlation between the HI and MN titres following LAIV administration. Specifically, the HI and MN responses against the A/H1N1 and B/Massachusetts strains exhibited a significant correlation in children and adults, whereas the A/H3N2 responses correlated only in the adult population. Correlations between HI and MN responses have been documented previously for IIVs [32]; the current study is one of a few to report such a correlation following LAIV administration [13]. Prior research has shown MN assays to be highly sensitive in measuring functional antibodies [33]. This finding is supported by MN data, where we observed detectable levels of MN antibodies against A/H3N2 and B/Massachusetts strains in >90% of vaccinees at days 28–56 after vaccination. Furthermore, we noted a fold increase in the MN response against the A/H1N1 strain at day 56 post-vaccination across all cohorts. This contrasts with our previous observation of lower induction of HI antibody response [16] and highlights a potential strength of the MN assay’s higher sensitivity in evaluating LAIV immunogenicity. 

Our previous findings showed that influenza–specific pre-existing antibodies gained through natural infection correlate negatively with LAIV-induced salivary IgA, tonsillar follicular helper T cells, memory B cells, and antibody secreting cell responses [16]. In contrast, data shown here indicate that the functional antibody responses are not significantly impacted by the vaccine recipient’s age, except for the NAI antibody responses against the B/Yamagata strain. We observed weaker B/Yamagata-specific NAI antibody responses in naive children than adults. This discrepancy may partly contribute to the increased disease severity of influenza B observed in children compared to adults [34]. The functional antibody responses offer a promising alternative for assessing LAIV immunogenicity. However, further clinical research is necessary to establish their use as a potential COP. 

Higher avidity has been correlated with increased protection from severe disease in live attenuated vaccines targeting cytomegalovirus and rubella, as well as inactivated influenza A/H1N1 vaccines [35,36,37,38]. In contrast, another study reporting on A/H1N1 pdm09 hospitalized patients correlated higher avidity with increased disease severity [39]. To our knowledge, the current study is the first to report the changes in antibody avidity following LAIV vaccination. The only significant change in avidity was detected in primed children less than 9 years of age in response to A/H1N1 HA. The primary infecting strain of this study group was likely to have been this 2009 pandemic A/H1N1 strain. In the study by Aartse et al. [40] it was shown that following primary infection, there is an initial increase in broadly reactive, low affinity HA stem-specific antibodies. This was followed by an elevation of higher affinity HA head-specific antibodies against the primary infecting strain in the blood over time through a process of affinity maturation in the germinal centres. The reasons for a lack of an increase in antibody avidity against A/H3N2 and B/Mass strains after LAIV are not clear. Thus, further investigations are needed to understand how infection and vaccination can influence the B cell memory and the antibody affinity maturation process. Such insights are crucial to understand how these factors shape the immune responses to future vaccinations. Further characterization of antibody subclasses and their changing avidity may also provide further insights into immunogenicity of LAIV. Limitations of this study relate to the small sample size reducing the statistical confidence of our results, and unequal gender distribution of cohorts as influenza immunogenicity has previously been shown to vary between male and females [41].

## 5. Conclusions

Overall, our data indicate that LAIV vaccination can elicit antibody responses in both naive and antigen-experienced adults and children. The kinetics and magnitude of this response vary depending on the assay used and the virus strain. This highlights the necessity of using a range of assays to assess LAIV vaccination, as the HI assay alone does not offer a comprehensive indication of its immunogenicity. 

## Figures and Tables

**Figure 1 vaccines-12-00864-f001:**
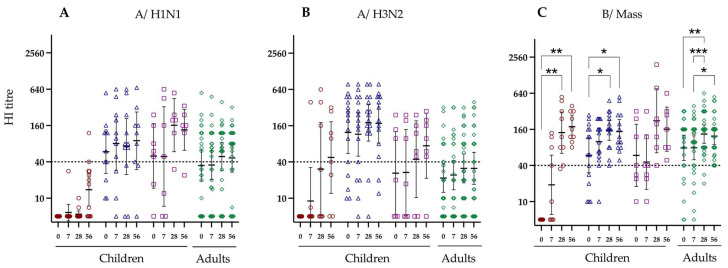
Hemagglutination inhibiting titres against the three vaccine strains in children stratified by virus exposure and adults. The virus strains are shown on the top of the figure and the Y-axis shows the HI titres, (**A**) A/California/7/2009 (pdm09 H1N1), (**B**) A/Texas/50/2012 (A/H3N2), (**C**) B/Massachusetts/2/2012. Each symbol indicates an individual HI response. Red circles indicate children under 9 years old who have not been primed, blue triangles represent children under 9 who have been primed, purple squares represent children above 9, and green diamonds represent adults. The dotted horizontal line at 40 refers to the protective HI titre. Significance levels calculated using Dunn’s test indicated as (*) *p* < 0.05, (**) *p* < 0.01, (***) *p* < 0.001.

**Figure 2 vaccines-12-00864-f002:**
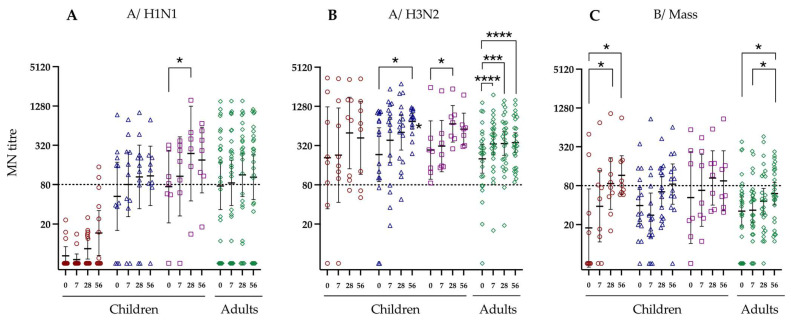
Microneutralization (MN) titres against the three vaccine strains in children stratified by influenza virus exposure and adults. The virus strains are shown on the top of the figure as (**A**) A/California/7/2009 (pdm09 H1N1), (**B**) A/Texas/50/2012 (A/H3N2), or (**C**) B/Massachusetts/2/2012, while the Y axis is the MN titre. Each symbol indicates an individual MN response. See Figure 1 for figure legends. The dotted horizontal line indicates the MN titre of 80. Horizontal bars indicate geometric mean titres for the group with 95% confidence intervals. Significance levels calculated using the Dunn’s test are indicated as (*) *p* < 0.05, (***) *p* < 0.001, (****) *p* < 0.0001.

**Figure 3 vaccines-12-00864-f003:**
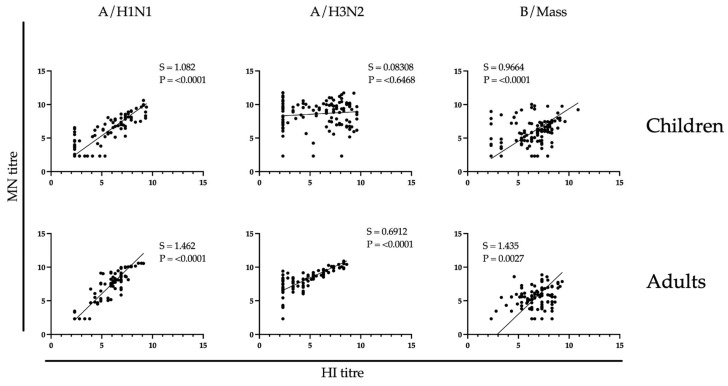
Regression analysis of microneutralization (MN) and hemagglutination inhibition (HI) antibody titres. The Deming regression analysis was performed using log_2_-transformed HI and MN antibody titres. The vaccine strains are indicated at the top of the figure as A/California/7/2009 (pdm09 H1N1), A/Texas/50/2012 (A/H3N2), or B/Massachusetts/2/2012. MN (*y*-axis) and HI (*x*-axis) titres from time points 0, day of tonsillectomy, 28-, and 56-days post-vaccination were plotted. The slope (S), and *p*-value (P) are shown for each correlation.

**Figure 4 vaccines-12-00864-f004:**
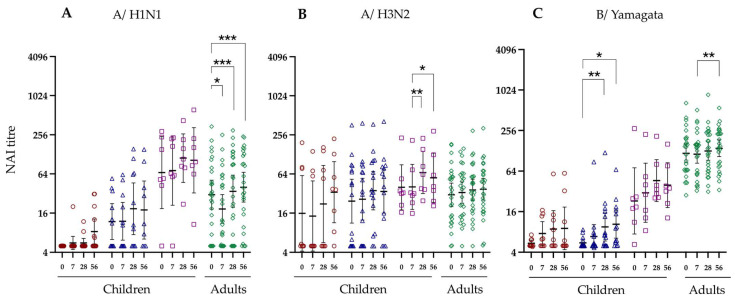
Neuraminidase inhibiting (NAI) antibody titres against the three vaccine strains in children were stratified by virus exposure and adults. The virus strains are shown on the top of the figure as (**A**) A/Equine/Prague/56 (H7) + A/California/07/09 (N1), (**B**) A/turkey/Massachusetts/3740/1965 (H6) + A/Texas750/2012 (N2), (**C**) A/turkey/Massachusetts/3740/1965 (H6) + B/Yamagata/16/1988 (NB), while the *Y*-axis is the NAI titre. Each point indicates an individual response and figure legends are presented in Figure 1. Horizontal bars indicate geometric mean titres for the group with 95% confidence intervals. Significance levels calculated using the Dunn’s test are indicated as (*) *p* < 0.05, (**) *p* < 0.01, (***) *p* < 0.001.

**Figure 5 vaccines-12-00864-f005:**
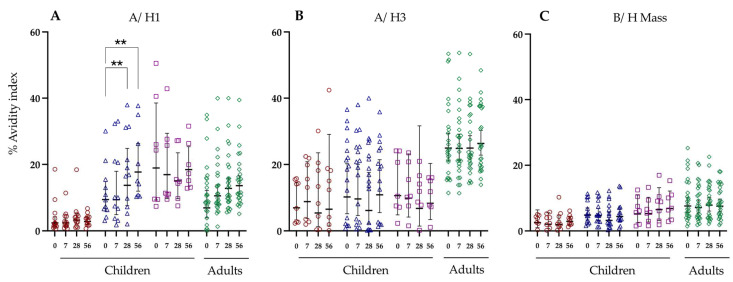
The antibody avidity to haemagglutinin in children stratified by virus exposure and adults. The hemagglutinin proteins are shown on the top of the figure (**A**) A/ H1, (**B**) A/ H3, (**C**) B/ H Mass, while the Y-axis is the avidity index percentage against Haemagglutinin (HA, top). Each symbol indicates an individual response. Red shows naive children below 9 years of age. Blue shows primed children below the age of 9 years. Purple shows primed children above the age of 9. Green shows the adult cohort. Horizontal bars indicate geometric mean titres for the group and 95% confidence intervals. Significance levels calculated using the Dunn’s test are indicated as (**) *p* < 0.01.

**Figure 6 vaccines-12-00864-f006:**
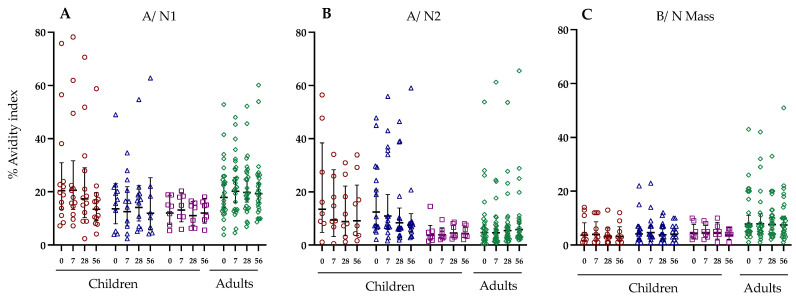
Neuraminidase avidity index changes against the three vaccine strains in children stratified by virus exposure and adults. The neuraminidase strains are shown on the top of the figure (**A**) A/ N1, (**B**) A/ N2, (**C**) B/ N Mass, while the Y-axis is the avidity index percentage against Neuraminidase (NA). Each symbol indicates an individual response. Red shows naive children below 9 years of age. Blue shows primed children below the age of 9 years. Purple shows primed children above the age of 9. Green shows the adult cohort. Horizontal bars indicate geometric mean titres for the group, and 95% confidence intervals.

**Figure 7 vaccines-12-00864-f007:**
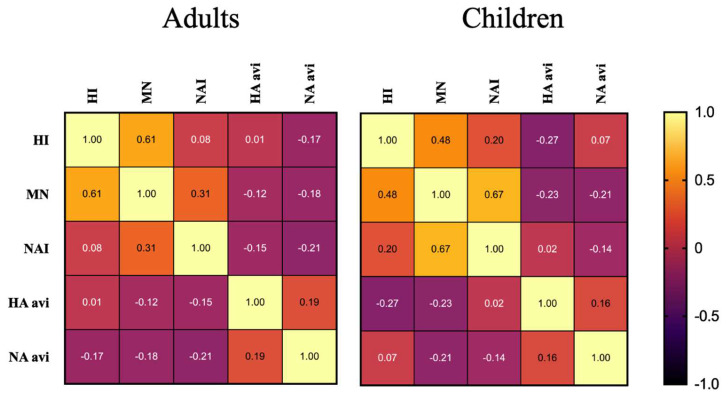
Assay Spearman R Correlation Matrix in Children and Adults Following LAIV. The fold-change between day 0 and 56 was taken for each vaccine strain, and then averaged to correlate changes between different assays.

**Table 1 vaccines-12-00864-t001:** The demographics of the subjects used in this study.

Characteristics				
Groups	Children under 9		Children 9 or above	Adults
*n*	24		7	26
Median Age (years)	4		13	29
Age Range (years)	3–7		11–17	18–59
Gender (%male)	67%		29%	27%
LAIV Doses	2		1	1
Previous Pandemic Vaccination *	25%		43%	50%
	Naïve	Primed	Age Primed	Age Primed
A/H1N1	13	11	7	26
A/H3N2	8	16		
B/Mass	9	15		

Upper panel shows the number of individuals (*n*), age, gender distribution, LAIV regiment, and prior vaccination of stratified cohorts assessed in this study. The lower panel describes the stratified cohort sizes according to vaccine strain assessed. Children under 9 were further stratified according to strain specific pre-vaccination HI titres, as either naïve ≤ 5 or primed > 5. * AS03 adjuvanted influenza A (H1N1) pdm09, (GSK) was administered during the 2009 pandemic.

## Data Availability

Data is available upon reasonable request to the senior corresponding author.

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
