# Peer review of "Impact of Pre-Existing Immunity and Age on Antibody Responses to Live Attenuated Influenza Vaccine"

_vaccines, 2024, doi:10.3390/vaccines12080864_

Round 1

Reviewer 1 Report

Comments and Suggestions for Authors

Thirty-one children and 26 adults in this clinical trial, the authors stratified cohort according to the number of LAIV doses (age), and pre-existing immunity in young children under the age of 9 years. The author also detected different antibody response by HI, MN and NAI et al..

The conclusion is not focused: positive correlation between HI and MN after LAIV vaccination; preexisting immunity and age impact Immunogenicity; MN is an alternative approach to assessing influenza vaccine immunogenicity and a potential COP.

Other comments:

1. All data are recommended to be supplemented with full age group analysis in addition to stratified analyses, which will allow for a better evaluation of the relevance of each test, as LAIV is a vaccine variety suitable for all ages.

2. Line 22, which is the standard correlate of protection for influenza vaccines. should be which is the standard correlate of protection for killed influenza vaccines.

3. The author should mention the efficacy of trivalent LAIV during the 2013-2014 influenza season.

4. Age impact vaccine immunogenicity may still be influenced by priming status.

5. The author should caculate fold increase for HAI, MN and NAI.

Reviewer 2 Report

Comments and Suggestions for Authors

Hoen et al.'s article, "Preexisting Immunity and Age Impact Live Attenuated Influenza Vaccine Immunogenicity," investigates how age and preexisting immunity affect the immunogenicity of live attenuated influenza vaccines (LAIV). The study includes 31 children and 26 adults immunized with trivalent LAIV during Norway's 2013-2014 influenza season. The primary objective is to understand the impact of these factors on eliciting functional and neutralizing antibody responses. In the present form, the article needs to correct several aspects:

Main concerns:

-        The title must highlight the main finding of the article, like “Impact of Preexisting Immunity and Age on Antibody Responses to Live Attenuated Influenza Vaccine.”

-        The abstract must incorporate the relevance by simply describing the significance of the poor HI response.

-        The Keywords must be improved by eliminating the technical and reducing the redundancy.

-        The introduction must provide a brief overview of the current understanding and gaps in knowledge regarding LAIV immunogenicity. Also, it must explain why understanding the impact of age and preexisting immunity is crucial. HA needs to be described. The article must include more citations to give robustness to the background.

-        In the method section, clarify the selection criteria for participants and any potential biases. Also, provide detailed descriptions of assay protocols.

-        Line 137: Include the citation of where the technique was modified.

-        Section 3.1 must be rewritten since it is too wordy.

-        The authors need to add a description of Table 1.

-        All the Figures are too small. The authors must change the position of the legend and enlarge it. The statistical methods used in this article are generally appropriate for the type of data and the study design. However, improvements can be made in the clarity and thoroughness of the statistical analysis section. Including a power analysis, confidence intervals, detailed descriptions of methods, handling of missing data, adjustments for multiple comparisons, and validation of assumptions will strengthen the statistical rigor of the study.

-        Line 257: include the respective citation.

-        The authors must highlight the most significant findings in the text to guide readers through the data.

-        In the discussion section, the authors must relate their findings to the broader context of influenza vaccine research. Also, it must Discuss any limitations of the study and how they might impact the results. It is essential to suggest future research directions based on the findings. More citations must be included to support their statements. The authors must emphasize how the findings could influence future vaccination strategies, particularly for different age groups.

Minor concerns:

-        Line 80: HA.

-        Line 101: remove (live attenuated influenza vaccine).

-        The authors must use the mentioned abbreviations during the text.

-        The grammar must be checked and improved.

-        Change all (fig. xx) for (Fig. xx).

-        Revise all the text and rewrite if needed to avoid wordiness.

Comments on the Quality of English Language

Moderate English editing is required.

Reviewer 3 Report

Comments and Suggestions for Authors

First, let me congratulate you on a well-designed study and an attempt to address a 'thorny ' issue.

I have several peripheral comments that you are not obliged to address but may spark some divergent thinking.

1. I have a recollection of reading two studies on influenza antibody profiles which I think were from a team at Cambridge UK. Are any of the concepts in those papers of relevance?

2. Any thoughts on the role of original antigenic sin in the antibody responses?

3. Regarding the figures, it has always occurred to me that many immunological studies should be better analyzed by attempting to determine if each set of data belongs to a different statistical distribution, i.e., log-normal, Weibull, etc, etc. Is such an approach possible?

4. Regarding the introduction you may wish to include a paragraph on the nonspecific effects of vaccines against all-cause mortality. Live attenuated vaccines are well recognized to give wider mortality protection than just against the target pathogen. Is there any possibility that one potential measure of protection may lie in this domain and if so which immune cells may be involved.

5. A study regarding the age profiles for deaths arising from different strains of COVID-19 observed that each strain had a unique single-year-of-age profile which was also gender specific. The authors raised the philosophical question as to whether each vaccine carried the same age specificity as the constituent strains. Such age specificity may have some form of antigenic distance to the currently circulating strains which may alter their protective capacity.

I hope that these thoughts may in some way help to contribute to the concepts in this paper.

Round 2

Reviewer 2 Report

Comments and Suggestions for Authors

The author has improved the article and responded to my suggestions. Only a few errors have to be corrected.

- Line 191: HA () or NA at 1 g/ml.

- All figures: Significances (*) are missing.

- Lines 462-464: Rewrite the sentence for clarity. Also, there are limitations, so avoid using "potential".

- The grammar of the Supplementary data must be reviewed and corrected.

Comments on the Quality of English Language

The quality is fine, but the grammar needs to be reviewed.
